# "It's his cheerfulness that gives me hope": A qualitative analysis of access to pediatric cancer care in Northern Tanzania

Madeline Metcalf[1]*, Happiness D. Kajoka[2], Esther Majaliwa[2,3], Anna Tupetz[1,4], Catherine A. Staton[1,4], João Ricardo Vissoci[1,4], Pamela Espinoza[1], Cesia Cotache-Condor[1,5], Henry E. Rice[1,6], Blandina T. Mmbaga[2,7], Emily R. Smith[1,4]

1 Duke Global Health Institute, Duke University, Durham, North Carolina, United States of America, 2 Kilimanjaro Christian Medical University College, Moshi, Tanzania, 3 Pediatric Hematology and Oncology Services, Kilimanjaro Christian Medical Centre, Moshi, Tanzania, 4 Department of Emergency Medicine, Duke University Medical Center, Durham, North Carolina, United States of America, 5 Duke Center for Global Surgery and Health Equity, Duke University, Durham, North Carolina, United States of America, 6 Division of Pediatric Surgery, Department of Surgery, Duke University Medical Center, Durham, North Carolina, United States of America, 7 Kilimanjaro Clinical Research Institute, Kilimanjaro Christian Medical Centre, Moshi, Tanzania

* madeline.metcalf@duke.edu

## Abstract

Pediatric cancer is a significant and growing burden in low- and middle-income countries. The objective of this project was to describe the factors influencing access to pediatric cancer care in Northern Tanzania using the Three Delays Model. This was a cross-sectional qualitative study conducted between June and August 2023 at Kilimanjaro Christian Medical Centre (KCMC). Using purposive sampling methods, caregivers of children obtaining pediatric cancer care at KCMC were approached for participation in in-depth interviews (IDIs) and a demographic survey. All IDIs were facilitated in Swahili by a bilingual research coordinator. Analysis utilized inductive and deductive coding approaches to identify dominant themes and sub-themes impacting access to pediatric oncology care. Data collection concluded once saturation was achieved at 13 IDIs, defined as the absence of new codes after three consecutive interviews. Participants reported significant financial barriers to accessing pediatric cancer care along the entire care continuum. In the first delay, themes included waiting for symptoms to resolve and the identification of initial symptoms. The most substantial delays occurred in delay 2, including health infrastructure at mid-level facilities, misdiagnoses, the referral system, travel, and traditional medicine. Participants did not describe delays after arrival to KCMC and rather offered perspective on their child's cancer diagnosis, their concerns while obtaining care, and their hopes for the future. Financial support provided by the Tanzanian government was the only facilitator noted by participants. We suggest targeted interventions including 1) empowerment of CHWs and local traditional healers to advocate for earlier care seeking behavior, 2) implementation of clinical structures and training at intermediary medical centers aimed at earlier referral to a treatment facility, 3) incorporation of support and education initiatives for families of children with a cancer diagnosis. Lastly, national health plans should include pediatric cancer care.

**Data Availability Statement:** Data are only available upon reasonable request and data transfer requires a written agreement approved by Kilimanjaro Christian Medical Centre Ethics Committee and the National Institute for Medical Research (Tanzania). Data inquiries can be sent to Gwamaka W. Nselela at gwamakawilliam14@gmail.com. A Data Transfer Agreement must be completed between the study investigators and the individual or entity requesting access to the study data before the data may be shared.

**Funding:** This project was funded by the Duke Global Health Institute Graduate Student funds (MM and PE), and the National Institutes of Health K01 Grant #5K01TW012181 (ERS). No other authors received specific funding for this work. The funders had no role in study design, data collection and analysis, decision to publish, or preparation of the manuscript.

**Competing interests:** The authors have declared that no competing interests exist.

## Introduction

The Lancet Oncology Commission on Sustainable Care for Children with Cancer estimates that there will be 13.7 million newly diagnosed cases of pediatric cancer between 2020 and 2050, nearly 90% of which will occur in low or lower-middle income countries (LMICs) [1]. Further, the Commission calculates that with the current global investment, diagnostic, and treatment provision trends, 11.1 million children will experience a cancer-attributable death during this 30 year period, with 84% of these children residing in LMICs [1]. Considering the World Health Organization (WHO) Global Initiative for Childhood Cancer's 2030 goal of 60% 5-year survival for pediatric cancer patients worldwide, significant scale-up in the financing, accessibility, and availability of quality pediatric oncology services is required [1–3].

In Tanzania, a LMIC in East Africa, several factors impact families' ability to obtain care for their child diagnosed with cancer such as health education, transportation, household finances, and traditional medicine [4, 5]. Tanzania has three health centers with the capacity to provide comprehensive cancer care to children, including diagnostic, laboratory, and treatment services. Six licensed pediatric oncologists serve the entire country with a population of more than 33 million children [6]. Due to workforce capacity challenges, training initiatives at Muhimbili University of Health and Allied Sciences have offered specialized training for nurses in both adult and pediatric oncology care. Despite these initiatives, research conducted at Kilimanjaro Christian Medical Centre (KCMC), one of the country's three tertiary level centers with cancer care capacity, reports that over 75% of pediatric cancer patients are diagnosed with stage III or IV cancer, suggesting notable delays in access to care [7].

Initially developed for examining maternal mortality in low-income settings, "Three Delays Model" has been used to classify barriers and facilitators to maternal, emergency, and neonatal care in Tanzania [4, 5, 8, 9]. However, the model has yet to be qualitatively applied to pediatric cancer care in the country. The Three Delays Model organizes barriers and facilitators to healthcare service procurement into three groups including: 1) delays in seeking care, 2) delays in reaching care, and 3) delays in receiving care [9]. The initial decision to seek care may be impacted by factors such as a household's socioeconomic status, health education, cultural or spiritual beliefs, and lack of geographic proximity or transportation to a health center [4]. For pediatric cancer, the second delay is associated with higher rates of morbidity and mortality and can be influenced by health service capacity and referral systems at smaller health centers [2, 10]. This second delay can also be impacted by similar sociocultural factors described in the first delay [4, 10]. The final delay is concerned with delays experienced once reaching the definitive medical center where diagnosis and treatment services can be provided. While not always contained within the framework, we have included financial influences within each of these delays to highlight the economic toll of pediatric cancer care.

Although there is a growing body of global epidemiological literature outlining the prevalence and incidence trends of pediatric cancer [7, 11, 12], there remains a gap in qualitative research that illustrates the experience of families in obtaining oncology care. To understand the factors influencing increased mortality and limited access to pediatric cancer care, we gathered caregiver perspectives of the journey to care with the Three Delays Model, beginning from the identification of initial symptoms and extending beyond obtaining care to include post-treatment plans.

## Methods

### Study design

This cross-sectional qualitative study used data collected from in-depth interviews (IDIs) with caregivers of children obtaining pediatric oncology care at KCMC between June (07/06/2023) and August (26/08/2023).

### Research team and reflexivity

The research team involved in data collection and analysis included a research coordinator (HK), KCMC's pediatric oncologist (EM), two graduate students (MM, PE), a postdoctoral fellow (AT) and an epidemiologist and principal investigator of the study (ERS). All IDIs were facilitated by the bilingual, female research coordinator (HK) based at KCMC and an employed member of the research team. The research coordinator is professionally trained as a medical doctor in Tanzania but was not involved in any patient care or known to the participants prior to enrollment. HK was trained in qualitative methods by AT prior to the commencement of the study. Before proceeding with the methods and findings from this study, the first and last author acknowledge their positionality as educated, White American women within an influential U.S. institution. Although actions were taken to support equitable partnership and limit bias, the authors recognize that this positionality may have inherently influenced the study design and interpretations of data.

### Setting

Located within the Kilimanjaro region of Tanzania, KCMC is classified as a tertiary level medical center, serving nearly 11 million patients each year [13]. The KCMC pediatric cancer ward serves an estimated 100–150 patients a year and currently hosts one oncologist (EM) who is responsible for the diagnosis and treatment of all pediatric cancer cases [14].

### Participant screening and enrollment

Using purposive sampling methods, potential participants were approached for study participation while presenting for care at the KCMC pediatric oncology ward between June and August 2023. In accordance with purposive sampling principles, KCMC's pediatric oncologist (EM) and the local research coordinator (HK) screened and enrolled participants to represent a wide range of cancer diagnoses, child ages, demographic characteristics, and experiences in obtaining cancer care. Potential participants were approached based on the following inclusion criteria: 1) patient was 14 years old or younger, 2) patient was actively obtaining treatment for a diagnosed cancer condition, 3) caregiver was at least 18 years old, 4) caregiver spoke Swahili or English (fluent languages of the research coordinator). Participants were excluded from the study if they had cognitive problems that would inhibit the caregiver's ability to participate or if the caregiver declined participation. Due to differences in treatment provision, inclusion was limited to patients between 0–14 years of age to report solely on children and early adolescents rather than older adolescents and young adults. During screening, two potential interviewees declined participation.

### Data collection procedures

All interviews took place in a quiet, private location in the KCMC pediatric oncology ward. The interviewer's name and role within the study were disclosed to participants prior to the interview. However, no additional information was shared regarding interviewer bias, assumptions, or personal interests in the study. After completing the written informed consent

procedures, the research coordinator completed a short paper survey with the participant to collect household demographic data such as relationship to the child, child's age and sex, urban/rural home community designation, ethnic and religious affiliation, education level, occupation, household and medical expenses, household monthly income, and health insurance status.

Interviews were performed using a semi-structured interview guide containing open-ended questions and probes based on the Three Delays Model. The interview guide was developed by our team (CCC, MM) based on the conceptual framework and an existing interview guide developed by the Harvard Medical School Program in Global Surgery and Social Change [10, 15]. We pilot tested the interview guide with two caregivers of children with a hemophilia disorder diagnosis and obtaining follow up services at KCMC. Based on the review and discussion of the pilot transcripts, final amendments were made to the guide. Mainly, questions were rephrased to be linguistically appropriate for the study population. The interview guide was developed in English and later translated to Swahili by the research coordinator (HK) and validated by the bilingual pediatric oncologist and co-principal investigator at KCMC (EM).

In addition to interviewer field notes, all interviews were audio recorded for transcription and translation. Field notes were not directly used in the final qualitative analysis, rather they helped frame and provide context regarding the findings. The research coordinator first transcribed interviews verbatim in Swahili and then translated the interviews into English. As an additional quality check, a KCMC research assistant not involved in the facilitation of the interviews compared the Swahili and English transcripts for translation validity. No repeat interviews were conducted, and transcripts were not returned to the participants for comments and/or corrections. Interviews lasted between 30–60 minutes and participants received no compensation for their participation. Audio recordings, Swahili interview transcripts, and final English transcript translations were uploaded to a password-protected and encrypted electronic data repository available to only the qualitative study team for analysis. Paper surveys were stored in a locked file cabinet separate from all informed consent and screening documents at KCMC until the demographic data had been de-identified and transferred to Duke University via the electronic data repository. Caregiver IDIs were conducted until the research team unanimously agreed that data saturation had been met, defined as the absence of new codes after three consecutive interviews.

## Analysis

After four interviews had been completed, the primary qualitative analyst (MM) coded the interviews using NVivo 12 software (Lumivero, Denver, CO, USA) [16]. Utilizing both inductive and deductive coding approaches, prevalent themes were identified within the Three Delays Model and a codebook was created to guide the analysis of the proceeding interviews. To maintain analytical validity, the codebook was discussed weekly between the analyst (MM), research coordinator (HK), and Duke University principal investigator (ERS). Changes were made to the identified themes and sub-themes as appropriate. No revisions were made after the 10th interview and all interviews were recoded with the final version of the codebook. For analytical rigor, a random sample of 6 interviews were coded by the KCMC research coordinator (HK) and a graduate research assistant (PE). For the triangulation of data, all analysts (MM, HK, PE) compared and discussed interpretations to ensure consistency and address any discrepancies in coding. A coding comparison query resulted in >90% coder agreeability between the three coders. This collaborative approach enhanced the validity and reliability of findings by incorporating diverse perspectives, limiting individual biases, and reinforcing identified themes.

### Ethical considerations

Written informed consent was obtained, and participants received a hard copy of the consent document in their preferred language (Swahili or English) prior to the start of the interview. Data storage and study procedures were approved by the Duke University and KCMC institutional review boards (IRBs) and the Tanzania National Institute for Medical Research (NIMR) prior to the commencement of the study. We utilized the Consolidated Criteria for Reporting Qualitative Research (COREQ) guidelines which is included as S1 Checklist with this manuscript [17].

## Results

### Demographic characteristics

Between June and August 2023, a total of 13 caregiver IDIs and complimentary demographic surveys were completed. Majority of the interviewed participants identified themselves as the child's mother (61.5%) and more than 90% of interviewees designated their household's home community as rural (Table 1). The most common household size including both adults and children living in the home was between 2–6 individuals (53.8%), with the most common religion being Christianity (53.8%). Of the 13 pediatric patients, 7 were male (53.8%) and 6 were female (46.2%). Three of these patients were between the ages of 0–2 years of age, four were between the ages of 3–6 years, two children were within 7–10 years, and four children were between the ages of 11–14 years. None of the interviewees reported having health insurance.

### Qualitative results

Organized within the Three Delays Model, Table 2 outlines the thematic codes included in this manuscript. In addition to outlining the delays experienced in seeking, reaching, and receiving care, participants provided contextual background regarding their journey to care at KCMC.

Analysis suggested a duality between delays discussed by IDI participants and delays perceived by the research team. IDI responses suggested that families experienced delays stemming from traditional medicine, finances (across all three delays), waiting for worsening symptoms, and hospital infrastructure. The qualitative research team agreed with all participant delays noted above, however, suggested the inclusion of referral systems and travel as additional barriers.

### Delay 1—Seeking care

**Initial symptoms.** Caregivers outlined the symptoms that motivated care seeking behavior. One participant detailed some of the symptoms noticed within their child before seeking care, *"At first, it looked like he had some sort of squinting or eye misalignment, but it wasn't squinting. It was actually his eye protruding because it was inside, and it started coming out."* (IDI Participant 6) Many participants suggested the initial local diagnosis *"child of the eye."*

Another respondent discussed the changes over time in the formation of their child's abdomen stating, *"I began to notice small lump appearing in the abdomen. As time went on, the lump grew larger, and the whole abdomen was distended."* (IDI Participant 8).

**Waiting for worsening symptoms.** Delays in initiation of care seeking behavior were often a result of waiting for the child's condition to worsen or resolve over time. One mother discussed how they delayed care seeking behavior for over six months with hope that the condition would heal. The mother stated, *"I thought it would go away on its own, that's why I waited because I had never seen anything like it before. So, I thought it would resolve itself, and I waited but rather it grew."* (IDI Participant 6).

**Table 1. Study population demographics.**

| Characteristic | Total |
|---|---|
| | % (n) 100.0 (13) |
| **Interviewee's Relationship to Child** | |
| Mother | 61.5 (8) |
| Father | 7.7 (1) |
| Sibling | 15.4 (2) |
| Uncle | 15.4 (2) |
| **Child's Age (Years)** | |
| 0–2 | 23.1 (3) |
| 3–6 | 30.8 (4) |
| 7–10 | 15.4 (2) |
| 11–14 | 30.8 (4) |
| **Child's Sex** | |
| Male | 53.8 (7) |
| Female | 46.2 (6) |
| **Rural-Urban Designation** | |
| Rural | 92.3 (12) |
| Urban | 7.7 (1) |
| **Household Size** | |
| 2–6 | 53.8 (7) |
| 7–11 | 38.5 (5) |
| 12+ | 7.7 (1) |
| **Caregiver Completed Education** | |
| No Formal Education | 7.7 (1) |
| Primary | 84.6 (11) |
| Secondary | 7.7 (1) |
| College | 0.0 (0) |
| **Religion** | |
| Christian | 53.8 (7) |
| Muslim | 30.8 (4) |
| None | 15.4 (2) |
| **Occupation of Main Household Provider** | |
| Skilled Employment | 7.7 (1) |
| Farming and Livestock | 69.2 (9) |
| Self Employed | 23.1 (3) |
| **Insurance Status** | |
| Yes | 0.0 (0) |
| No | 100.0 (13) |

Another participant explained how their child's condition recurred despite medicine provided at home, *"I would give him medicine, and he would seem better, so he would go back to school. It went on like that, back and forth. He would get better, then worse. But then, he got really sick, all of his nostrils were like rotten. The sores were like rotting wounds."* (IDI Participant 12).

**Finances.** More than half of participants discussed how household finances resulted in an early delay to seeking care. Caregivers explained how they had to sell land or possessions to initiate care seeking. A sibling describes this within their household, *"The trees they sold were*

**Table 2. Thematic codes presented in this manuscript.**

| Three Delays Model | | |
|---|---|---|
| **Seeking Care** | **Reaching Care** | **Receiving Care** |
| *Delays*:<br>• Waiting for Worsening Symptoms<br>• Finances | *Delays*:<br>• Traditional Medicine<br>• Hospital Infrastructure<br>• Referral System<br>• Travel<br>• Finances | *Delays*:<br>• Finances |
| *Contextual Factors*:<br>• Initial Symptoms | *Contextual Factors*:<br>• Care Received Prior to KCMC | *Contextual Factors*:<br>• Care Received at KCMC<br>• Caregiver Concerns<br>• Treatment Success |

*trees they had planted on their property, and when they faced financial difficulties, they decided to sell them to get money for treatment.*" (IDI Participant 10) Another participant described how they had to *"throw"* or sell possessions to collect enough money for transportation to the first medical center. *"I had to carry a bag of rice and throw it so that we could get transportation to the healthcare center. . .I had some roosters, and I threw them, and that's what helped me to travel to Ifakara."* (IDI Participant 2).

A participant noted how government support has served as a financial facilitator in seeking care for their child. *"We were financially constrained, and when the child fell ill, we didn't have the means to seek immediate medical attention. Fortunately, the government stepped in and brought me here. . .If it weren't for the government's support, I don't know what would have happened."* (IDI Participant 1) No additional examples of government or social support schemes for financing medical expenses were provided.

## Delay 2—Reaching care

**Traditional medicine.** Once the care seeking process began, cultural and spiritual beliefs suggesting traditional medicine techniques influenced when caregivers sought care from health centers. Many participants noted how traditional medicine was used as first aid before opting to seek additional medical care. One participant explained how traditional remedies have proven effective within their community, *"Yes, there are some people who have used them and found them helpful. . . But for me, they didn't help."* (IDI Participant 1).

Participants also discussed how economic factors impacted their decision to use traditional medicines, often noting the *"fear"* associated with the costs of medical treatment, particularly in rural areas (IDI Participant 2). *"The other factor that has contributed is finances because you find the child falls suddenly ill and you don't have anything in your pocket, you say, let's try traditional medicine since even other children in the village have used it and recovered."* (IDI Participant 3).

**Care received prior to KCMC.** Many participants explained how their child's condition had been misinterpreted at intermediary health facilities and been prescribed ineffective medications prior to KCMC. One mother explained how their child had repeatedly been treated for malaria before obtaining their cancer diagnosis, *"So, whenever he had a fever at night, we would rush him to the hospital. They would say it's just a fever or sometimes say he has severe malaria."* (IDI Participant 1) Most often, children were prescribed pain medications which failed to provide long-term relief or treatment for the child's condition. *"It was about two*

*months that he was using those medications. Then I went to Ifakara, and they also gave me some medications just for soothing."* (IDI Participant 2).

A minority of respondents discussed how initial testing was done at smaller health facilities which led to their final arrival at KCMC, but diagnostic services were never mentioned at these medical centers. One participant explained this experience, *"When we reached the regional hospital, they tested him for liver disease, took an X-ray, did an ultrasound, but the medication didn't help either. That's when they told us to come here (KCMC)."* (IDI Participant 3).

**Hospital infrastructure.**   More than half of respondents suggested that workforce, laboratory, diagnostic, and treatment capacity for pediatric patients at smaller health centers contributed to delays in arrival to KCMC. One mother explained the experience of not being able to obtain testing services for their child at multiple health facilities, *"When you go to a hospital, they just touch the child briefly and say they can't help, or they just look at him and say they can't do anything, go somewhere else. You move to another hospital, and they say the same thing. Meanwhile, the child's condition is troubling, and he's crying. As a mother, you feel bad, and sometimes you even start crying out of pity for your child. . . They just look at the child without conducting proper tests."* (IDI Participant 1).

Another respondent explained their perspective regarding the capacity to provide quality and sufficient care at some medical centers and how this impacts the child's health status, *"Well, you know, when I started at those smaller hospitals, what I realized is that you could have a patient that is okay and later on their disease worsens just because they lack that specific expertise or equipment."* (IDI Participant 11).

**Referral system.**   Respondents described how they were referred to multiple mid-level facilities before KCMC, outlining how the stepwise referral system within the country contributed to delays. While these referrals may have been linked to other delays in diagnoses and infrastructure outlined above, the system itself presented challenges in obtaining timely care. One participant explained the challenges related to delayed referrals, *"Perhaps one challenge is that when you arrive at a hospital, they should give you a referral early because sometimes you arrive, and they perform tests and tests. You find you have spent like 400,000 shillings and the child has not even improved. Later on, when you're already broke that's when they give you a referral."* (IDI Participant 3).

Despite participants commonly noting multiple week- or month-long delays in being referred to KCMC, caregivers expressed overall satisfaction with the referral system, focusing on how the system had helped their family. *"I haven't seen any issue because this is where the treatment is. Even if I had continued to want to remain there and there was no treatment so when they brought me here, it wasn't a problem for me because I saw and believed that my child would get better here."* (IDI Participant 8).

**Travel.**   Caregivers and children endured long journeys to receive care at KCMC. Travel time varied between participants, ranging from three hours to two days. One participant explained the emotional and physical challenges of traveling with a sick child, *"It was difficult for me to travel with the child in public transport the wound was still fresh, and I was carrying the child in my arms."* (IDI Participant 4) Caregivers also expressed variability in the travel experience depending on the type of transportation and familiarity with the area, *"I traveled alone. I didn't know Moshi and I didn't know anyone here. . .I arrived in Moshi at 2 o'clock at night! When I got here at 2am I didn't have any familiarity, I didn't know anyone and there was nobody there. They dropped me off along the way. I sat with the child on the roadside."* (IDI Participant 5).

**Finances.**   For the second delay, finances played a key role in the provision of care prior to reaching KCMC. Households were generally required to pay out-of-pocket (OOP) for all medical expenses related to their child's illness at intermediary hospitals before obtaining a KCMC

referral. A participant explained how financial concerns of paying for medical care prohibited them from attending the clinic stating, *"There was a month when I couldn't go. They told me I needed to bring 240,000 shillings for the child to undergo a test. The date arrived, but I hadn't received the money.."* (IDI Participant 8).

Similar to the first delay, more than half of participants described how they had to sell possessions or borrow money to continue seeking care. An interviewee explained how they both borrowed money and sold livestock to pay for medical expenses, *"I borrowed money from someone, and when I returned home, I sold some items at home to pay them back. I also sold some of my cows to continue with my child's treatment."* (IDI Participant 4) Another participant suggested how households can end up with *"nothing"* after selling these possessions, *"If you have cows, you sell them. If you have something, you sell it. It all ends, and you find that nothing is left."* (IDI Participant 12).

### Delay 3—Receiving care

**Care received at KCMC.**   Participants discussed the care they had received at KCMC, often describing how the child was diagnosed and their perspective of what treatment was being provided. One participant outlined the initial care provided by KCMC clinical staff, *"they started running tests on his head, and the next day, they told me that he needed surgery to prevent it from spreading to the other eye. He had the surgery, and I stayed there for three days before coming here to the cancer ward."* (IDI Participant 6).

Although most caregivers opted to begin treatment immediately for their child at KCMC, potential delays presented for families who returned home before initiating treatment. One mother describes her experience after obtaining her child's cancer diagnosis, *"When we came here to KCMC in July, the eye was not that swollen, and the eye had not come out and the examination revealed the cancer. . . After they gave me the results, I went back home. . .I tried covering the eye that had no disease, and I gave the child something to look at, but I realized he couldn't see it nor anything else. That's when I made up my mind to come back."* (IDI Participant 5).

**Caregiver concerns.**   Participants described some of their worries about being away from home such as familial relationships, household economic implications of obtaining care, and the health trajectory of their child. A participant describes their worries about the additional children at home and how being away from home may be impacting the children's school attendance, *"Yes, worrying is a must because we left our young children at home. They go to school on their own, and if I'm not there, they might only go for two or three days a week, or sometimes they don't go at all."* (IDI Participant 3) Another respondent expressed the sense of hopelessness that can accompany a cancer diagnosis. *"There is no worse illness than this. You need great support. I see people here laughing and appearing well-fed, and I feel like something is wrong in their brains this is not a place to laugh; it's not a safe place. I'd prefer being in jail than here because in jail, you'll eventually get out alive, but here, it's by grace."* (IDI Participant 12).

Nearly half of participants described how they drew upon their religious or spiritual beliefs to manage their worries and find a sense of comfort and *"grace"* (IDI Participant 11). One participant stated, *"I just pray to God, rely on him, and ask God for strength. When I pray and lean on God, the fear diminishes."* (IDI Participant 4) Further, a child's uncle suggests how the pride they feel in reaching care has lessened their concerns, *"I don't have any worries because I have saved a life. I only think about the child I have saved."* (IDI Participant 2).

**Treatment success.**   Despite concerns, participants expressed a sense of hope in their perceived effectiveness of the treatment provided at KCMC. Nearly all participants felt like their child would complete the treatment plan, rarely noting additional delays that may prevent them continuing to receive services. Caregivers felt reassured by the progress in their child's

condition, which motivated them to continue with the treatment plan. One respondent said, *"just seeing the child playing gives me some peace of mind."* (IDI Participant 5) Another participant expressed their satisfaction with KCMC as a medical facility compared to another hospital, *"KCMC is the biggest and the best treatment here. . . So, if anyone has a problem, I will tell them to directly come to KCMC for salvation."* (IDI Participant 2).

**Finances.** Finally, participants described how finances impacted their ability to continue receiving oncology care and avoid treatment abandonment. When asked if they experienced any stress regarding paying for oncology care, one mother noted how the philanthropic financing available at KCMC allows her to not have financial worries stating, *"Not at all, because they told me that the child's treatment costs here at the hospital are covered. They told me there are no expenses, my cost is my time."* (IDI Participant 7) Participants expressed a sense of relief in knowing they would be able to continue treatment for their child without incurring catastrophic medical expenses. However, before arriving to KCMC, respondents were unaware of this financing option and often noted how the perceived price of treatment in Moshi was *"incomprehensible"* (IDI Participant 2). One participant said, *"I knew that it might involve a lot of money because we are used to hearing that when you hear about KCMC, you know you need to be well-prepared. They say the tests themselves are expensive, and just entering the hospital itself is costly, unlike other hospitals."* (IDI Participant 3).

While this did not necessarily cause a delay in the continuation of care, participants discussed how they had to forego their normal employment or livelihood activities regardless of the free treatment provided. One mother explained how the lost income will impact her family, *"It affects us significantly because my job was a great source of income. . . It's the biggest challenge that I see affecting our lives."* (IDI Participant 4).

## Discussion

To the best of our knowledge, this is the first paper to qualitatively apply the Three Delays Model to pediatric cancer care in Northern Tanzania. We report the most substantial delays occur once families have sought care and experience barriers related to the Tanzanian referral system and health infrastructure capacity. While global health research has pushed for decentralized healthcare expansion in LMICs, health system and workforce capacity must concurrently advance to support the appropriate delivery of pediatric cancer care. Overall, an increase in community and clinical resources aimed at reducing financial constraints and ensuring quality, timely access to pediatric oncology care is required in Tanzania. We recommend that future interventions extend beyond basic access to care to include holistic strategies that aid families in completing full treatment regimes.

Respondents described initial delays due to waiting for the child's condition to resolve or deteriorate before seeking medical care. Although caregivers expressed how they identified symptoms and changes in the child's behavior, unlike injurious or emergency conditions, early signs of pediatric cancer can be difficult to discern. Due to the complexities of recognizing signs of pediatric cancer and the lack of specialized oncology services in rural communities, we recommend that community health workers (CHWs) and traditional healers be trained to identify and suggest when further medical treatment should be sought. A study conducted in Cameroon developed an early recognition and referral training for primary healthcare workers, including traditional healers, and found that healthcare workers knowledge of early childhood cancer symptoms increased by 52% after completing the training program [18]. The WHO Global Strategy on Human Resources for Health further suggests the value of CHWs, and urges countries to embrace CHWs as diverse and sustainable primary healthcare professionals [19, 20]. Building on the current literature and global recommendations, future

interventions should engage CHWs and traditional healers in cost-effective and culturally reflective pediatric cancer training and outreach, thus limiting delays at the household level.

Out of all the barriers mentioned, health system barriers were the most often discussed among respondents. While the stepwise referral system in Tanzania has benefits for preventing the treatment of mild conditions at high-level facilities, our study suggests that the referral system hindered patients from obtaining care in a timely manner. Keating et al. report that referral delays for severe pediatric illnesses put children at greater risk for mortality in Tanzania, which we believe to be applicable to pediatric cancer [4]. Additionally, the costs associated with misdiagnosis and ineffective treatments were considerable and prohibited some participants from accessing referrals promptly. While caregiver's did not perceive this referral system to have hindered the overall treatment or prognosis of their child, the global pediatric oncology literature strongly supports the necessity of prompt diagnosis for treatment effectiveness, minimizing out-of-pocket (OOP) costs, and reducing the suffering accompanying a cancer diagnosis [21]. Households in LMICs, such as Tanzania, are more likely to finance medical expenses by selling assets, using personal savings, and borrowing money, impacting the long-term socioeconomic status of the household [22]. These financial strategies are supported by the data we present and are exacerbated by the lack of health insurance in our study sample. Smith et al. has highlighted the cost-effectiveness, economic growth, and equity building that investment in global pediatric surgical care produces, principles relevant to global pediatric cancer care [23]. Progressive financial protection schemes such as individual-level support for financing indirect medical expenses such as lodging, food, and transportation and macro-level interventions such as the inclusion of pediatric oncology diagnosis and treatment in universal health coverage (UHC) plans must be considered. In addition, it will be critical that the capacity to recognize children with potential cancer conditions improves at smaller health facilities. As a zonal referral hospital, KCMC has built a collaboration with Princess Máxima Center for Pediatric Oncology to provide biannual, specialized pediatric oncology education to all KCMC oncology staff. To minimize disease progression and household OOPs costs, similar collaborations or training initiatives should be considered at smaller, community health facilities. Workers within these facilities should be trained to identify concerning symptoms of pediatric cancer and refer these patients to a definitive treatment facility. Efforts addressing these concerns have proven successful in Tanzania with Muhimbili National Hospital reporting an 8% treatment abandonment rate for acute leukemia after accommodation was provided for families of children obtaining cancer care in addition to infrastructure advancements [24, 25].

While the severity of cancer was well known, respondents expressed varying thoughts surrounding the causes of cancer, how cancer may reoccur, treatment strategies, and follow up care. Participants also suggested the importance of spirituality and support networks for coping with their child's diagnosis. The feelings described by participants represent the necessity of clear health education and tangible strategies for managing the emotions that accompany a cancer diagnosis. The National Cancer Institute (NIC) has developed six elements of patient-centered communication in cancer care including 1) fostering healing relationships, 2) responding to emotions, 3) exchanging information, 4) making decisions, 5) managing uncertainty, and 6) enabling self-management [26]. Although these elements have mostly been applied to HICs, a review has recommended the NCI guidelines as a baseline for further research and intervention in LMICs [27]. Cancer support groups have been implemented in numerous settings to support families in HICs. However, similar programs have yet to be widely incorporated in LMICs. Participant responses detailing worries about being away from home and the confusion that accompanied their child's cancer diagnosis suggests that future childhood cancer interventions could focus on delivering comprehensible health education and providing supportive spaces for managing fears and uncertainty. Due to treatment

abandonment concerns, this extension of health education is especially valuable in pediatric oncology settings where routine follow-up care is necessary.

While these results will inform future interventions aimed at increasing the quality and accessibility of pediatric cancer care in Tanzania, our study contains important limitations. Like any qualitative research study, these findings are not intended to be representative of the entire pediatric oncology patient population. Rather, this study intends to detail the journey to pediatric cancer care and identify potential opportunities for intervention that support timely and quality service provision. Second, although the thematic codebook underwent routine discussion and adaptation based on the perspectives of research team members based at KCMC, not all interviews were doubled coded, and agreeability was determined by a random sample of 6 interviews. While this is a notable limitation, no major discrepancies in coding outcomes were noted. between Lastly, we report perspectives of only households who received care at KCMC, excluding the perspectives of families who were unsuccessful in reaching a final destination for oncology treatment and therefore may have experienced more significant barriers to accessing care.

## Conclusion

In this qualitative study, we report on the journey to pediatric cancer care beginning at the household level and extending beyond care received at KCMC to describe stressors and plans for the future. Our recommendations include 1) empowerment of CHWs and local traditional healers to advocate for earlier care seeking behavior at the community level, 2) implementation of financial protection, clinical structures, and training at intermediary medical centers aimed at earlier referral to a definitive treatment facility, 3) incorporation of support and health education initiatives for families of children with a cancer diagnosis. Most importantly, in accordance with the United Nation's target of achieving universal health coverage (UHC) worldwide by 2030 [28], future national health plans should include pediatric cancer care.

## Supporting information

**S1 Checklist. COREQ checklist.**
(DOCX)

**S2 Checklist. Inclusivity in global research.**
(DOCX)

## Author Contributions

**Conceptualization:** Madeline Metcalf, Pamela Espinoza, Cesia Cotache-Condor, Henry E. Rice, Emily R. Smith.

**Data curation:** Madeline Metcalf, Happiness D. Kajoka, Esther Majaliwa, Emily R. Smith.

**Formal analysis:** Madeline Metcalf, Happiness D. Kajoka, Anna Tupetz, Pamela Espinoza, Emily R. Smith.

**Funding acquisition:** Esther Majaliwa, Catherine A. Staton, Blandina T. Mmbaga, Emily R. Smith.

**Investigation:** Madeline Metcalf, Happiness D. Kajoka, Esther Majaliwa, Anna Tupetz, Pamela Espinoza, Emily R. Smith.

**Methodology:** Madeline Metcalf, Anna Tupetz, Catherine A. Staton, Pamela Espinoza, Emily R. Smith.

**Project administration:** Madeline Metcalf, Esther Majaliwa, Catherine A. Staton, Emily R. Smith.

**Resources:** Catherine A. Staton, Emily R. Smith.

**Supervision:** Anna Tupetz, Catherine A. Staton, Blandina T. Mmbaga, Emily R. Smith.

**Writing – original draft:** Madeline Metcalf, Emily R. Smith.

**Writing – review & editing:** Madeline Metcalf, Happiness D. Kajoka, Esther Majaliwa, Anna Tupetz, Catherine A. Staton, João Ricardo Vissoci, Pamela Espinoza, Cesia Cotache-Condor, Henry E. Rice, Emily R. Smith.

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
