## [Decision Letter · Decision Letter 0]

19 Sep 2024

PGPH-D-24-01495

“It’s his cheerfulness that gives me hope”: A Qualitative Analysis of Access to Pediatric Cancer Care in Northern Tanzania

Dear Dr. Metcalf,

Thank you for submitting your manuscript to PLOS Global Public Health. After careful consideration, we feel that it has merit but does not fully meet PLOS Global Public Health’s publication criteria as it currently stands. Therefore, we invite you to submit a revised version of the manuscript that addresses the points raised during the review process.

The manuscript has been assessed by two reviewers and their comments are available below. They have requested more detailed methodology among other comments. Please review their comments and make all of the appropriate revisions. 

We look forward to receiving your revised manuscript.

Kind regards,

Emma Campbell, Ph.D

Staff Editor

Journal Requirements:

Reviewers' comments:

Reviewer's Responses to Questions

**Comments to the Author**

1. Does this manuscript meet PLOS Global Public Health’s publication criteria? Is the manuscript technically sound, and do the data support the conclusions? The manuscript must describe methodologically and ethically rigorous research with conclusions that are appropriately drawn based on the data presented.

Reviewer #1: Yes

Reviewer #2: Yes

2. Has the statistical analysis been performed appropriately and rigorously?

Reviewer #1: Yes

Reviewer #2: N/A

3. Have the authors made all data underlying the findings in their manuscript fully available (please refer to the Data Availability Statement at the start of the manuscript PDF file)?

Reviewer #1: No

Reviewer #2: Yes

4. Is the manuscript presented in an intelligible fashion and written in standard English?

Reviewer #1: Yes

Reviewer #2: Yes

5. Review Comments to the Author

Reviewer #1: This is an interesting manuscript utilizing a model introduced for ObstetRics to describe a common problem in LMICs. The manuscript gives an interesting post-mortem of the problem of delayed treatment from the standpoint of the dynamics within the healthcare system in Tanzania. The limitation of the study is the near uniformity of the participants and the fact that they were mostly low socio-economic class families. It is a well thought out and executed research endeavor.

Reviewer #2: Dear Authors,

Congratulations on your work! This is an interesting and a relevant study. I added below some suggestions:

Abstract:

• Sampling and sample size were not clearly described, e.g., purposive sampling and saturation with 13 IDIs. This should be aligned with the methods section in the main text.

• In the results section, I wonder if you could list the themes clearly.

Introduction:

• A great highlight was included about the capacity of pediatric oncology services in Tanzania. Well done. I wonder if you could elaborate a bit and describe if you had other challenges like trained pediatric oncology nurses; and how the local health care system deals with the demand on the pediatric oncology services by such a large number of newly diagnosed patients annually.

• A minor suggestion, consider removing the linking words, like “Further”

Method

• It is great to read about the research team and reflexivity. I wonder if you could add the team positionality.

• For purposive sampling, I wonder if you could elaborate a bit and describe how you achieved the maximum variation of the sample.

• Could you elaborate a bit and describe how the field notes were used in the analysis? How about triangulation?

Discussion

• I wonder if you could link the lack of local resources (e.g., 1 oncologist for 100 to 150 patients a year, and the lack of trained pediatric oncology nurses) with the reported factors and how that would lead to delays in diagnosis/treatment access and treatment abandonment after the child start receiving treatment.

6. PLOS authors have the option to publish the peer review history of their article (what does this mean?). If published, this will include your full peer review and any attached files.

**Do you want your identity to be public for this peer review?** For information about this choice, including consent withdrawal, please see our Privacy Policy.

Reviewer #1: No

Reviewer #2: No

---

## [Decision Letter · Decision Letter 1]

7 Nov 2024

“It’s his cheerfulness that gives me hope”: A Qualitative Analysis of Access to Pediatric Cancer Care in Northern Tanzania

PGPH-D-24-01495R1

Dear Ms. Metcalf,

We are pleased to inform you that your manuscript '“It’s his cheerfulness that gives me hope”: A Qualitative Analysis of Access to Pediatric Cancer Care in Northern Tanzania' has been provisionally accepted for publication in PLOS Global Public Health.

Best regards,

Sarah E. Brewer, PhD

Academic Editor

Reviewer Comments (if any, and for reference):

Reviewer's Responses to Questions

**Comments to the Author**

1. If the authors have adequately addressed your comments raised in a previous round of review and you feel that this manuscript is now acceptable for publication, you may indicate that here to bypass the “Comments to the Author” section, enter your conflict of interest statement in the “Confidential to Editor” section, and submit your "Accept" recommendation.

Reviewer #2: All comments have been addressed

2. Does this manuscript meet PLOS Global Public Health’s publication criteria? Is the manuscript technically sound, and do the data support the conclusions? The manuscript must describe methodologically and ethically rigorous research with conclusions that are appropriately drawn based on the data presented.

Reviewer #2: Yes

3. Has the statistical analysis been performed appropriately and rigorously?

Reviewer #2: N/A

4. Have the authors made all data underlying the findings in their manuscript fully available (please refer to the Data Availability Statement at the start of the manuscript PDF file)?

Reviewer #2: Yes

5. Is the manuscript presented in an intelligible fashion and written in standard English?

Reviewer #2: Yes

6. Review Comments to the Author

Reviewer #2: The authors have done an excellent job addressing all the comments and suggestions. No further revisions are necessary. Congratulations on your work!

7. PLOS authors have the option to publish the peer review history of their article (what does this mean?). If published, this will include your full peer review and any attached files.

**Do you want your identity to be public for this peer review?** For information about this choice, including consent withdrawal, please see our Privacy Policy.

Reviewer #2: **Yes: **Mohammad R. Alqudimat
